# Genome-wide association analysis and Mendelian randomization proteomics identify drug targets for heart failure

Danielle Rasooly [1,2,30] ✉, Gina M. Peloso [2,3,30], Alexandre C. Pereira[4,5], Hesam Dashti[1,6], Claudia Giambartolomei [7,8], Eleanor Wheeler [9], Nay Aung [10,11], Brian R. Ferolito[2], Maik Pietzner [9,12,13], Eric H. Farber-Eger [14], Quinn Stanton Wells[15], Nicole M. Kosik [2], Liam Gaziano[2,16], Daniel C. Posner[2], A. Patrícia Bento[17], Qin Hui[18,19], Chang Liu [18], Krishna Aragam [2,6,20], Zeyuan Wang[18], Brian Charest[2], Jennifer E. Huffman [2], Peter W. F. Wilson[19,21], Lawrence S. Phillips[19,22], John Whittaker [23], Patricia B. Munroe [24,25], Steffen E. Petersen [11,26], Kelly Cho[1,2], Andrew R. Leach [17], María Paula Magariños[17], John Michael Gaziano[1,2], VA Million Veteran Program*, Claudia Langenberg[9,12,13,31], Yan V. Sun [18,19,27,31], Jacob Joseph[28,29,31] ✉ & Juan P. Casas[1,2,31]

We conduct a large-scale meta-analysis of heart failure genome-wide association studies (GWAS) consisting of over 90,000 heart failure cases and more than 1 million control individuals of European ancestry to uncover novel genetic determinants for heart failure. Using the GWAS results and blood protein quantitative loci, we perform Mendelian randomization and colocalization analyses on human proteins to provide putative causal evidence for the role of druggable proteins in the genesis of heart failure. We identify 39 genome-wide significant heart failure risk variants, of which 18 are previously unreported. Using a combination of Mendelian randomization proteomics and genetic cis-only colocalization analyses, we identify 10 additional putatively causal genes for heart failure. Findings from GWAS and Mendelian randomization-proteomics identify seven (*CAMK2D*, *PRKD1*, *PRKD3*, *MAPK3*, *TNFSF12*, *APOC3* and *NAE1*) proteins as potential targets for interventions to be used in primary prevention of heart failure.

Heart failure (HF) is one of the most important threats to the sustainability of health systems in the United States[1]. Despite major improvements in the understanding of risk factors for incident HF[2], this knowledge has not yet been fully translated into effective interventions for the primary prevention of HF, except for blood pressure (BP) lowering medications[3] and statins[4]. Due to the inherent attributes of human genetics that minimize the risk of residual confounding and reverse causation[5], large-scale genomic analyses provide an opportunity to uncover putative causal mechanisms for complex phenotypes such as HF[6]. Recent genome-wide association studies (GWAS) of HF by the Heart Failure Molecular Epidemiology for Therapeutic Targets (HERMES) and the Million Veteran Program (MVP)[7] have identified 26 genomic loci associated with HF[8]. This emerging knowledge has served to identify novel biological mechanisms associated with incident HF and may inform the development of novel interventions for the primary prevention of HF.

A full list of affiliations appears at the end of the paper. *A list of authors and their affiliations appears at the end of the paper.

✉e-mail: drasooly@bwh.harvard.edu; jacob.joseph@va.gov

Novel technological developments can simultaneously measure thousands of human proteins in a single blood sample. The SOMAscan V4 assay includes 5207 aptamers capable of measuring 4988 unique human proteins, of which 514 are the target of drugs licensed or in the clinical phase, 1153 are the target of compounds in the preclinical phase, and 1377 are proteins predicted to be druggable[9,10]. This offers a unique opportunity for translating the genomic findings of HF into novel interventions for the primary prevention of HF. Given that human proteins account for the majority of targets for approved drugs to date and that expression or activity is central to the development of human disease[11], leveraging GWAS data of HF and protein quantitative trait loci (pQTL) offers an opportunity to provide mechanistic insight into the causal pathways involved in the emergence of HF as well as to inform novel therapeutic targets.

Here, we conduct a meta-analysis of GWAS on HF from the MVP and the HERMES consortium and leverage our GWAS of HF with pQTLs from the Fenland study to conduct Mendelian randomization (MR) and genetic colocalization analyses on human proteins covered by SOMAscan V4[12]. We then perform extensive downstream analyses covering HF risk factors, cardiac MRI traits, -omics, and downstream transcriptomics analyses to investigate the biological credibility of our genetic findings.

## Results

### Genome-wide meta-analysis identifies 18 novel loci for HF

We meta-analyzed GWAS on HF from the HERMES consortium and MVP (Supplementary Data 1) and identified variants at GW-significance ($p < 5 \times 10^{-8}$) (Fig. 1). The quantile-quantile (Q-Q) plot of the meta-analysis is shown in Supplementary Fig. 1. We performed follow-up analysis of the newly discovered HF variants to identify the likely causal gene for each signal and to investigate associations with 15 HF risk factors and nine left ventricular (LV) cardiac MRI traits.

We performed meta-analyses of genome-wide association results for HF from two studies: MVP ($n_{cases} = 43,344$; $n_{controls} = 258,943$) and HERMES ($n_{cases} = 47,309$; $n_{controls} = 930,014$). After quality control, we obtained association results for 10,227,138 genetic variants with HF. We observed 39 variants with genome-wide significant signals with HF, of which 18 variants were >500KB from a previously reported indexed variant (Fig. 2 and Supplementary Data 2). We performed fine-mapping using GWAS summary statistics (Supplementary Fig. 2). We determined the gene closest to the indexed SNP, as well as the gene with the highest score from Polygenic Priority Score (PoPs)[13] within a 500KB region of the indexed SNP (Table 1). PoPs take genome-wide features into account while the nearest gene is based on local information, providing complementary information for annotation of indexed

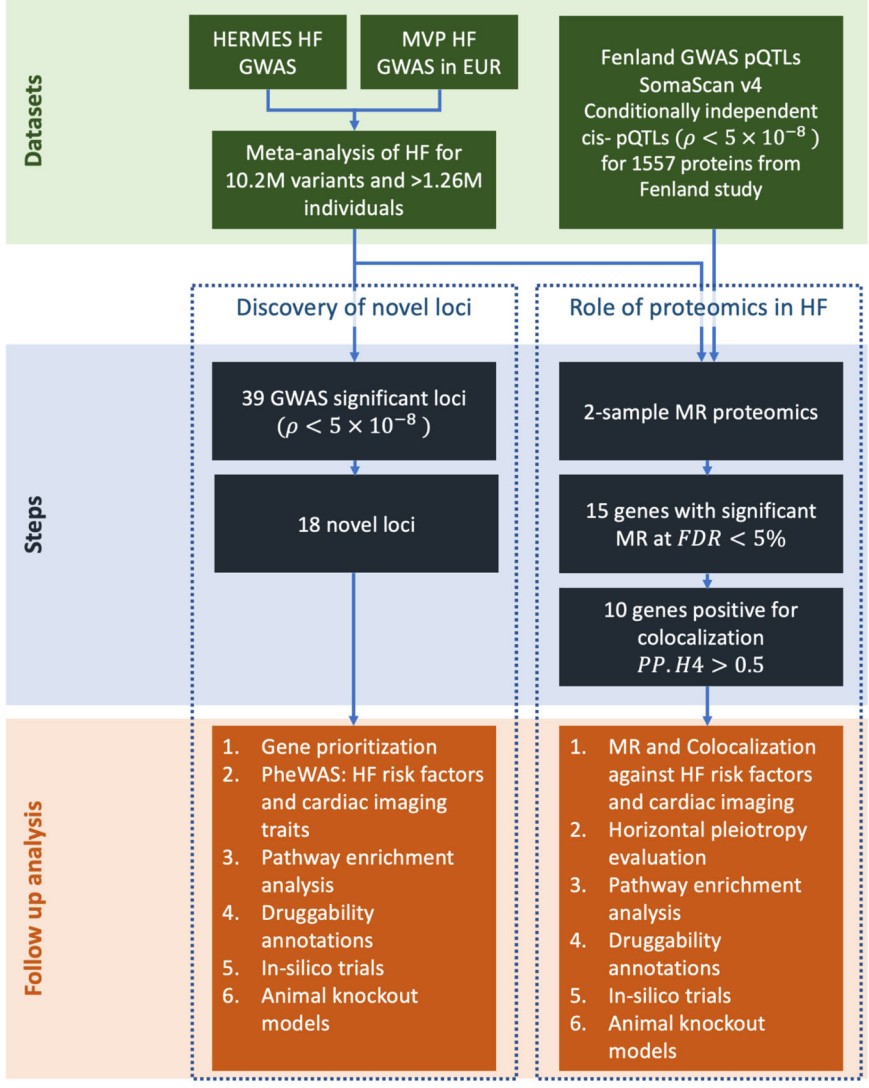

**Fig. 1 | Schematic diagram of the datasets and analyses.** HF heart failure, MVP Million Veteran Program cohort, GWAS genome-wide association study, pQTL protein quantitative trait loci, PheWAS phenome-wide association study, MR Mendelian randomization, FDR false discovery rate, PP.H4 posterior probability of $H_4$.

**(a)**

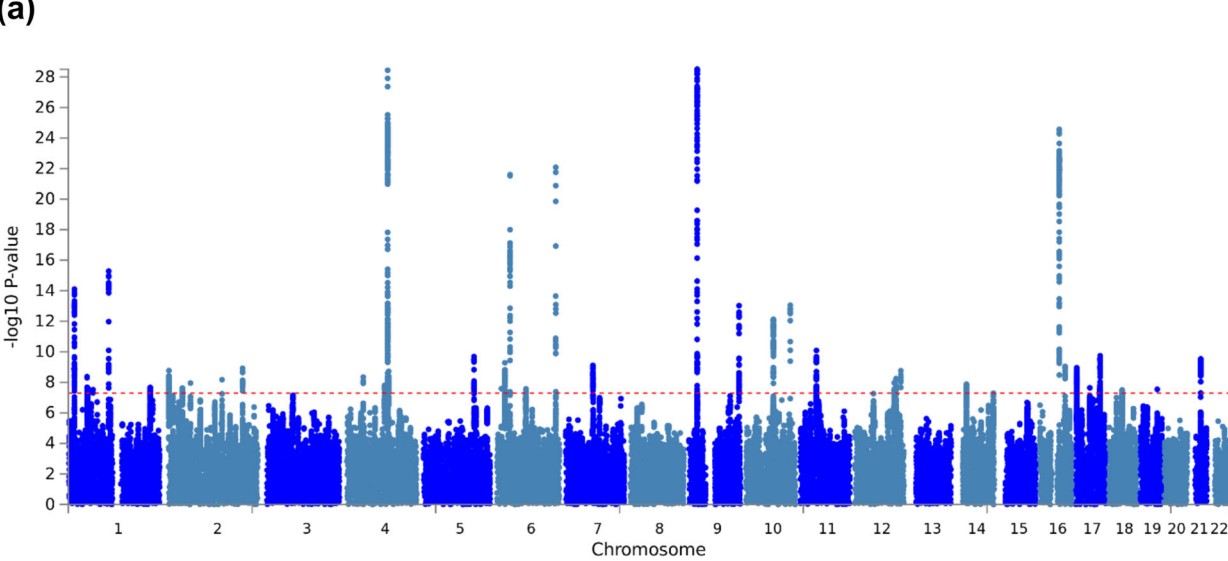

**(b)**

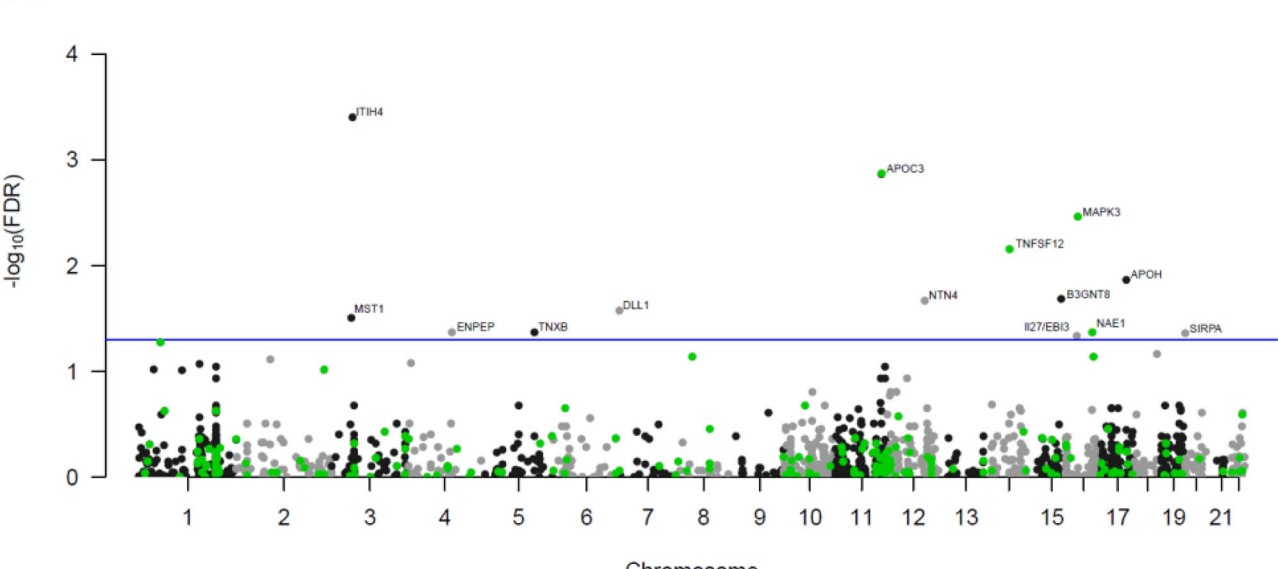

**Fig. 2 | Manhattan plots showing associations with HF from a GWAS meta-analysis on *n* = 1,266,315 individuals and b MR-wide proteomics. a** Manhattan plot showing the −log₁₀(*P* value) of association for each SNP from the GWAS meta-analysis plotted on the y-axis against genomic position on the x-axis. The red dotted line corresponds to the genome-wide significance threshold. The summary statistics of independent lead SNPs are noted in Supplementary Data 1. **b** Manhattan plot showing the −log₁₀-transformed FDR-adjusted *P* value of association for each gene plotted against genomic position on the x-axis. All tests were two-sided and adjusted for multiple comparisons. The blue line corresponds to an FDR threshold of 5% and points are color-coded by drug tractability information based on data provided by OpenTargets; green for druggable genes. FDR false discovery rate.

variants (see Methods). For all the genes suggested by the nearest gene and PoPS, we retrieved the results from gene-burden tests using putative Loss-of-Function (pLoF) variants from the Genebass-UK Biobank resource (see Methods)[14]. *RFX4* and *UBC*, both suggested by PoPs, showed the most significant gene-based *p* values with HF (*p* values of $9.12 \times 10^{-4}$ and $4.6 \times 10^{-3}$, respectively). From herein, we used genes suggested by PoPs as default to describe the distinct variants.

Except for rs6945340/*HIP1* and rs79682748/*SGIP1*, all other distinct variants for HF had an association (defined as 0.01/number of secondary traits, $p < 1 \times 10^{-4}$) with at least one HF risk factor (Fig. 3a). Five variants had the largest number of associations with HF risk factors: rs9352691/*PHIP* (blood pressure, body mass index (BMI), high-density lipoprotein cholesterol (HDL-C), alcohol consumption, and

atrial fibrillation (AF)), rs12992672/*TMEM18* (BMI, HDL-C, type-2 diabetes mellitus (T2DM), AF, and smoking), rs4755720/ *HSD17B12* (BMI, HDL-C, T2DM, and CAD), rs233806/*BANK1* (blood pressure, HDL-C, and BMI) and rs959388/*PRKD1* (BMI, smoking, and blood pressure), details in Supplementary Data 3. We observed that the directionality of the associations with HF risk factors was concordant with the findings on HF risk in 32 out of the 42 (76%) associations. HDL-C and diastolic BP accounted for nine of the ten discordant associations (Supplementary Fig. 3). We did not find associations with troponin, NT-proBNP, and IL-6 (Supplementary Data 3).

Only three variants (rs3820888/*SPATS2L*, rs4755720/*HSD17B12*, and rs72688573/*FAF1)* showed at least one association ($p < 1 \times 10^{-4}$) with LV cardiac MRI traits (Supplementary Fig. 4 and Supplementary

**Table 1 | Loci reported for HF in the meta-analysis of HERMES and MVP HF GWAS datasets**

| rsID | Chr | Pos | Nearest Gene | PoPs gene | CADD Phred Score | NEA | EA | MAF | Beta | SE | *P* value |
|------|-----|-----|--------------|-----------|------------------|-----|----|----|------|----|----------|
| *Novel variants* | | | | | | | | | | | |
| rs4755720 | 11 | 43628749 | *HSD17B12* | *HSD17B12* | 7.526 | C | T | 0.400 | −0.037 | 0.006 | 8.14E-11 |
| rs7766436 | 6 | 22598259 | *HDGFL1* | *HDGFL1* | 1.902 | C | T | 0.286 | 0.037 | 0.006 | 5.18E-10 |
| rs3820888 | 2 | 201180023 | *SPATS2L* | *SPATS2L* | 6.261 | C | T | 0.378 | −0.034 | 0.006 | 1.20E-09 |
| rs12992672 | 2 | 632592 | *TMEM18* | *TMEM18* | 1.061 | G | A | 0.172 | 0.045 | 0.007 | 1.70E-09 |
| rs10846742 | 12 | 125308682 | *SCARB1* | *UBC* | 0.277 | G | A | 0.173 | −0.046 | 0.008 | 1.73E-09 |
| rs17620390 | 4 | 114384328 | ***CAMK2D*** | ***CAMK2D*** | 1.907 | C | A | 0.284 | −0.037 | 0.006 | 1.90E-09 |
| rs72688573 | 1 | 50746997 | *FAF1* | *FAF1* | 6.834 | C | T | 0.022 | −0.122 | 0.021 | 4.31E-09 |
| rs10938398 | 4 | 45186139 | *GNPDA2* | *N/A* | 1.094 | G | A | 0.421 | 0.033 | 0.006 | 4.50E-09 |
| rs6945340 | 7 | 75100124 | *POM121C* | *HIP1* | 3.58 | C | T | 0.208 | −0.040 | 0.007 | 5.89E-09 |
| rs7564469 | 2 | 145258445 | *ZEB2* | *GTDC1* | 19.56 | C | T | 0.165 | −0.043 | 0.007 | 6.66E-09 |
| rs7977247 | 12 | 107259470 | *RIC8B* | *RFX4* | 1.868 | C | T | 0.434 | 0.032 | 0.006 | 1.07E-08 |
| rs1016287 | 2 | 59305625 | *FANCL* | *N/A* | 19.22 | C | T | 0.280 | 0.037 | 0.006 | 1.11E-08 |
| rs959388 | 14 | 30169987 | ***PRKD1*** | ***PRKD1*** | 0.596 | G | T | 0.417 | −0.031 | 0.006 | 1.30E-08 |
| rs233806 | 4 | 103212846 | *SLC39A8* | *BANK1* | 9.713 | C | T | 0.207 | −0.037 | 0.007 | 1.57E-08 |
| rs17038861 | 2 | 37233265 | *HEATR5B* | ***PRKD3*** | 0.403 | G | T | 0.195 | 0.039 | 0.007 | 2.35E-08 |
| rs9352691 | 6 | 79785607 | *PHIP* | *PHIP* | 6.373 | C | T | 0.365 | 0.032 | 0.006 | 2.65E-08 |
| rs10520390 | 19 | 46327831 | *SYMPK* | *DMWD* | 2.849 | G | C | 0.059 | 0.074 | 0.013 | 2.87E-08 |
| rs79682748 | 1 | 66989719 | *SGIP1* | *SGIP1* | 4.719 | G | A | 0.018 | −0.155 | 0.028 | 3.00E-08 |
| *Previously reported variants* | | | | | | | | | | | |
| rs7859727 | 9 | 22102165 | *CDKN2B* | *CDKN2A* | 1.448 | C | T | 0.488 | 0.061 | 0.006 | 3.11E-29 |
| rs2634071 | 4 | 111669220 | *PITX2* | *PITX2* | 1.622 | C | T | 0.219 | 0.079 | 0.007 | 3.64E-29 |
| rs11642015 | 16 | 53802494 | *FTO* | *RPGRIP1L* | 4.826 | C | T | 0.432 | 0.058 | 0.006 | 2.69E-25 |
| rs10455872 | 6 | 161010118 | *LPA* | ***PLG*** | 0.146 | G | A | 0.074 | −0.104 | 0.011 | 8.20E-23 |
| rs3176326 | 6 | 36647289 | *CDKN1A* | *CDKN1A* | 10.88 | G | A | 0.173 | −0.068 | 0.007 | 2.51E-22 |
| rs602633 | 1 | 109821511 | *PSRC1* | *CELSR2* | 8.63 | G | T | 0.207 | −0.054 | 0.007 | 5.19E-16 |
| rs1739833 | 1 | 16331108 | *C1orf64* | *ZBTB17* | 4.205 | C | T | 0.331 | −0.048 | 0.006 | 7.89E-15 |
| rs17617337 | 10 | 121426884 | *BAG3* | *BAG3* | 0.079 | C | T | 0.218 | −0.050 | 0.007 | 8.88E-14 |
| rs600038 | 9 | 136151806 | *ABO* | *SURF1* | 7.596 | C | T | 0.215 | −0.049 | 0.007 | 9.44E-14 |
| rs34163229 | 10 | 75406912 | *SYNPO2L* | *SEC24C* | 24 | G | T | 0.133 | −0.056 | 0.008 | 7.40E-13 |
| rs113437066 | 17 | 65836220 | *BPTF* | *BPTF* | 2.588 | ATTT | A | 0.197 | 0.061 | 0.010 | 1.81E-10 |
| rs11746435 | 5 | 137006762 | *KLHL3* | *HNRNPA0* | 7.216 | T | A | 0.229 | 0.042 | 0.007 | 2.04E-10 |
| rs2832275 | 21 | 30602994 | *BACH1* | *LTN1* | 1.08 | T | A | 0.139 | −0.047 | 0.008 | 2.90E-10 |
| rs7795282 | 7 | 74122857 | *GTF2I* | *GTF2IRD1* | 0.262 | G | A | 0.221 | −0.042 | 0.007 | 7.69E-10 |
| rs12933292 | 16 | 69566309 | *NFAT5* | *NFAT5* | 0.403 | G | C | 0.425 | 0.034 | 0.006 | 8.96E-10 |
| rs216199 | 17 | 2200871 | *SMG6* | *SMG6* | 3.442 | C | T | 0.388 | −0.037 | 0.006 | 1.11E-09 |
| rs2013002 | 12 | 112200150 | ***ALDH2*** | *ATXN2* | 4.336 | C | T | 0.415 | 0.033 | 0.006 | 5.68E-09 |
| rs17163345 | 1 | 222806218 | *MIA3* | *MIA3* | 7.541 | G | A | 0.270 | −0.034 | 0.006 | 2.15E-08 |
| rs3764351 | 17 | 37824339 | *PNMT* | *MED1* | 5.182 | G | A | 0.340 | −0.033 | 0.006 | 2.27E-08 |
| rs9349379 | 6 | 12903957 | *PHACTR1* | *PHACTR1* | 5.478 | G | A | 0.401 | −0.031 | 0.006 | 2.58E-08 |
| rs4327120 | 18 | 36532976 | *N/A* | *N/A* | 1.032 | C | T | 0.128 | 0.050 | 0.009 | 3.09E-08 |

Findings were identified using fixed effects inverse-variance weighted meta-analysis. The chromosomal position is based on GRCh37/hg19 reference. Gene names are italicized. Genes that are druggable or predicted to be druggable are highlighted in bold.
*CADD* combined annotation-dependent depletion, *NEA* non-effect allele, *EA* effect allele, *MAF* minor allele frequency, *SE* standard error.

Data 3). The rs3820888/*SPATS2L* variant was associated with six LV cardiac MRI traits and AF; all these associations were directionally concordant with the HF findings. The rs4755720/*HSD17B12* variant was associated with LV end-diastolic volume indexed to body surface area and four HF risk factors, and rs72688573/*FAF1* was associated with LV mass to end-diastolic volume ratio and two HF risk factors, see details in Supplementary Data 3. In the African-American subpopulation from the MVP GWAS (Supplementary Data 4), we found none of our 39 genome-wide significant distinct variants with HF in the European datasets achieved genome-wide significance (Supplementary Data 5).

**MR Proteomics and colocalization identifies ten genes for HF**
Using the GWAS data on SOMAscan V4 proteomics, we selected conditionally independent *cis*-variants, defined as any variant within a +/− 1 Mb region of the protein-encoding gene, that is associated with plasma levels of SOMAscan proteins ($p < 5 \times 10^{-8}$). We propose that these variants are instrumental variables for measured SOMAscan proteins and conducted two-sample MR analyses using our European-descent GWAS meta-analysis of HF from the MVP and HERMES consortium. We conducted several analyses to minimize confounding and biases. For the MR results that passed our

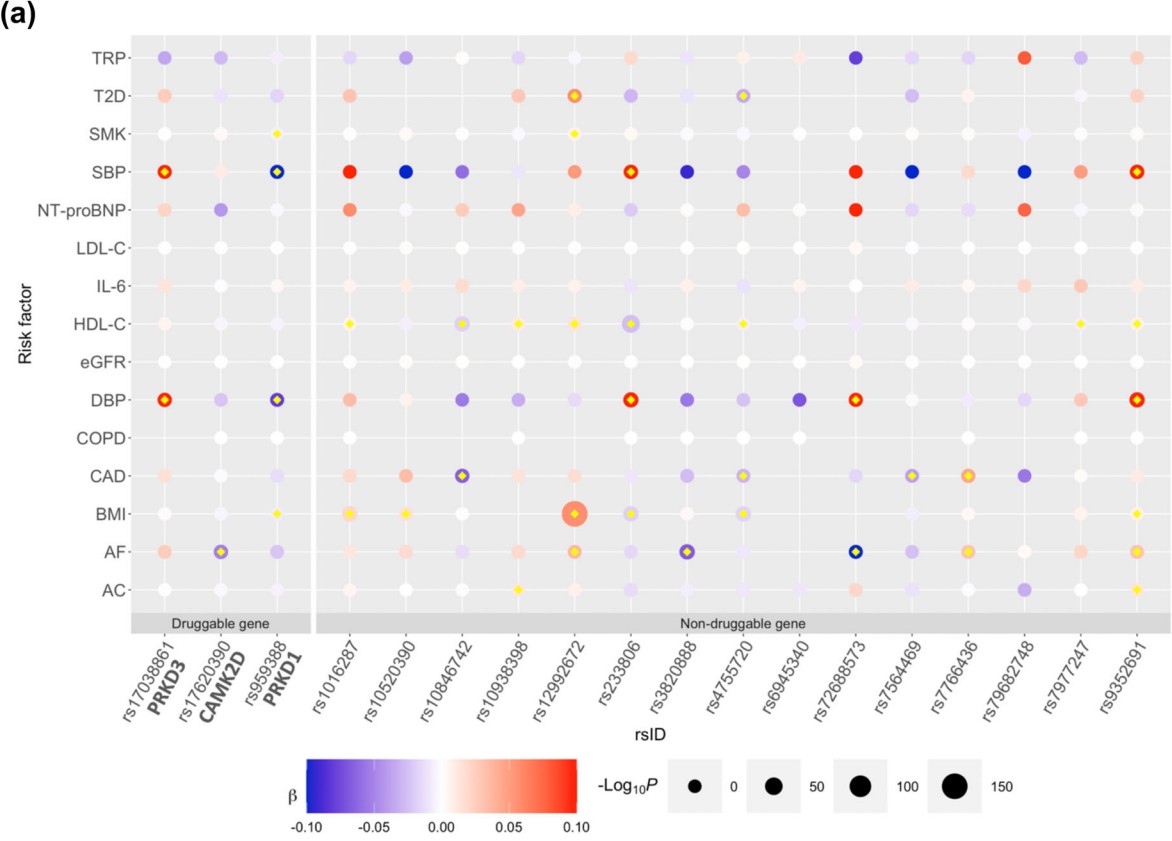

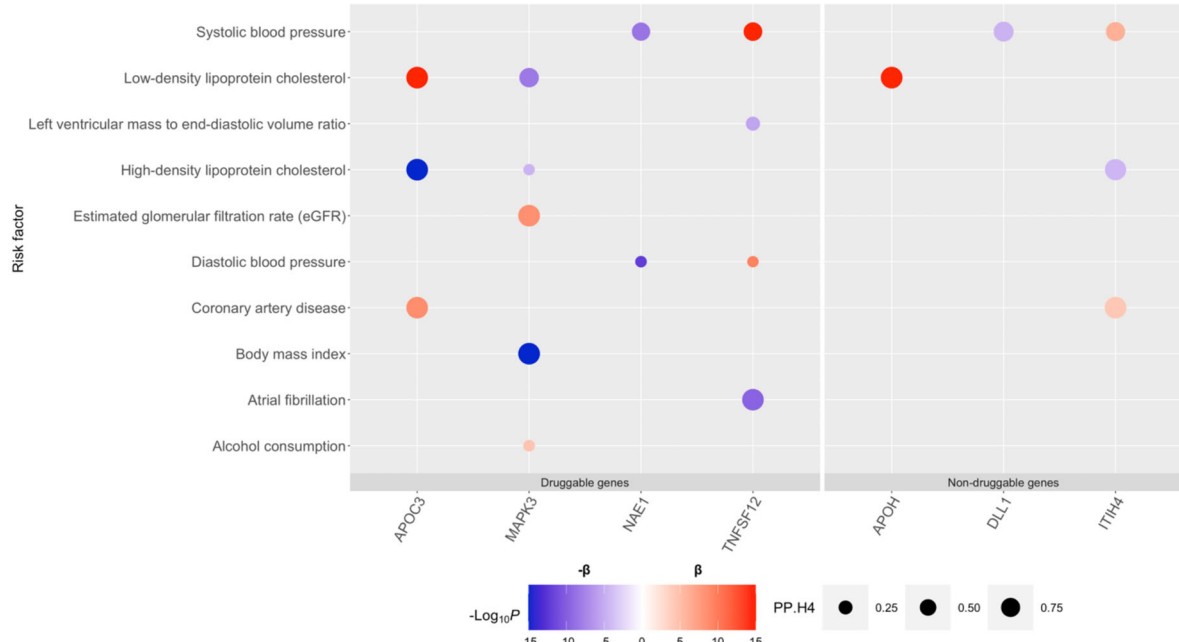

significance threshold (FDR <5%), we performed genetic colocalization analysis to ensure the MR results were unlikely to be confounded by linkage disequilibrium (LD). For the MR results with evidence of colocalization, we conducted MR and colocalization analyses against HF risk factors and cardiac MRI traits and *cis*-eQTL searches. Then, we conducted a novel multi-step analytical approach to reduce the risk of horizontal pleiotropy.

We used 2900 *cis*-pQTLs across 1557 genes from the Fenland study as proposed instrumental variables for conducting two-sample MR of proteomics with HF. We found 16 genes passed our MR threshold (FDR <5%), of which ten genes also showed suggestive evidence of colocalization between HF and pQTL signals (posterior probably of Hypothesis 4 (PP.H4): one common causal variant >0.5) for at least one of the instruments, and of which three genes show strong

**Fig. 3 | Plots showing a genetic association of 18 HF loci against risk factors for HF and b MR and colocalization estimates of MR-proteomic genes-hits against HF risk factors. a** The color of the bubble corresponds to the beta coefficient of the genetic association between the loci (x-axis) and trait (y-axis). Blue corresponds to a negative and red corresponds to a positive beta coefficient. The size of each bubble corresponds to the negative logarithm of the association $p$ value; larger size corresponds to lower $p$ values. Loci are grouped by druggable and non-druggable genes. All tests were two-sided without adjustment for multiple comparisons. Associations which passed the $p$ value threshold ($p < 1 \times 10^{-4}$) are denoted by a yellow diamond. **b** This bubble plot shows MR estimates for which $p < 1 \times 10^{-4}$. The size of each bubble corresponds to the posterior probability for hypothesis 4 derived from colocalization. The color of the bubble corresponds to the beta coefficient derived from MR. Blue corresponds to a negative association and red corresponds to a positive association; note that a positive $\beta$ indicates either an increase in protein levels corresponding to an increase in HF risk or a decrease in protein levels corresponding to a decrease in HF risk, while a negative $\beta$ indicates either a decrease in protein levels corresponding to an increase in HF risk or an increase in protein levels corresponding to a decrease in HF risk. The intensity of the color corresponds to $-\log_{10}(P \text{ value})$ for the strength of association in the MR. All tests were two-sided without adjustment for multiple comparisons. Loci are grouped by druggable and non-druggable genes. *TNXB*, *SIRPA*, and *ENPEP* genes are not included as these had no MR estimates on HF risk factors that pass the $p < 1 \times 10^{-4}$ threshold. $\beta$, Beta coefficient, AC alcohol consumption, AF atrial fibrillation, BMI body mass index, CAD coronary artery disease, COPD chronic obstructive pulmonary disease, DBP diastolic blood pressure, eGFR estimated glomerular filtration rate, HDL-C high-density lipoprotein cholesterol, IL-6 Interleukin-6, LDL-C low-density lipoprotein cholesterol, NT-proBNP N-terminal proBNP, SBP systolic blood pressure, SMK smoking, T2D type-2 diabetes, TRP troponin I cardiac muscle, PP.H4 posterior probability of $H_4$.

evidence of colocalization (PP.H4 >0.8), see details in Table 2 and Supplementary Data 6. Except for *ENPEP*, no other gene that colocalized was within 500 KB of a known HF GWAS loci. For genes with more than one instrument, we did not observe any evidence of heterogeneity based on Cochran's Q statistic according to the IVW model or by the MR-Egger intercept test, Table 2. This lack of heterogeneity suggests that average directional horizontal pleiotropy may not explain these findings.

Except for *ENPEP*, *TNXB*, and *SIRPA*, all the other genes that passed thresholds for MR and colocalization with HF also showed an association (defined as MR $p < 1 \times 10^{-4}$ and colocalization: PP.H4 >0.5) with at least one of the 15 HF risk factors (Fig. 3b and Supplementary Data 7). We observed that the directionality of the MR associations with HF risk factors was concordant with the MR findings on HF in 10 out of the 14 (71%) associations. HDL-C, LDL-C, and systolic BP accounted for discordant associations. Only the *TNFSF12* gene showed an association with an LV cardiac MRI trait that passed statistical thresholds for MR and colocalization, see details in Supplementary Data 7, 8. We investigated if the *cis*-pQTL instruments for the ten MR genes were also *cis*-eQTLs ($p < 5 \times 10^{-8}$). Twelve of the 18 proposed instruments were also *cis*-eQTLs in at least one tissue. None of the *cis*-pQTLs used as proposed instruments for *TNXB*, *APOC3*, and *APOH* genes showed a *cis*-eQTL association (Supplementary Data 9).

In our assessment of horizontal pleiotropy (see Methods and Supplementary Fig. 5), the 18 proposed instruments for the ten MR genes were associated ($p < 5 \times 10^{-8}$) with 251 proteins or gene expression using SOMAscan V4, Fenland study and eQTLGen, respectively (Supplementary Data 10). For 217 of the 251 proteins/gene expression, we identified at least one *cis*-pQTL or *cis*-eQTL at $p < 5 \times 10^{-8}$ associated with protein levels based on the SOMAscan V4 Fenland study, or gene expression based on eQTLGen. We then conducted two-sample MR of these secondary proteins/genes expression against HF and identified four genes (*TP53*, *ZNF259*, *ACVR2A, and MYRF*) that passed multiple testing thresholds (0.05/217, $p < 2 \times 10^{-4}$, Supplementary Data 10). These four secondary genes correspond to the following genes identified by MR proteomics as hits for HF: *TNXB* (*ACVR2A* and *MYRF*), *APOH* (*TP53*), and *APOC3* (*ZNF259*). *TP53* and *ACVR2A* were in a different biological pathway than *APOH* and *TNXB*, respectively, suggesting potential horizontal pleiotropy. *ZNF259* and *MYRF* did not retrieve any biological pathways; hence, it is unknown if these are due to horizontal pleiotropy. We then determined protein–protein interaction (PPI) networks for APOH and TNXB proteins using Enrichr and GPS-Prot databases. The Enrichr's PPI Hub Protein pathways reported interactions between APOH and CDC42, AKT1, TP53, and GRB2 (adjusted $p$ values <0.04), while the GPS-Prot showed that the APOH protein is directly connected to TP53 with confidence >0.6 (Supplementary Fig. 6). No significant interaction was identified for the TNXB and ACVR2A proteins.

## Genetic correlation estimates
Estimates of the genetic correlation between HF and 15 HF risk factors are reported in Supplementary Data 11. Results that pass multiple testing at 5% FDR are denoted, including a positive genetic correlation between HF and BMI of 0.56 (0.03) and with AF of 0.11 (0.02), as well as a negative genetic correlation between HF and HDL-C of −0.36 (0.03) (Supplementary Data 11).

## Polygenic risk score validation
To test the PRS for HF in an out-of-sample cohort, we used data from 75,119 participants of European descent from the BioVU, of which 5845 participants had HF. Individuals with a 1-standard deviation increase in the PRS had a 1.28 higher odds of HF (95% confidence interval (CI), 1.24–1.31; $p < 2 \times 10^{-16}$). Participants in the top decile had a 1.82-fold (95% CI, 1.60–2.06; $p < 0.0001$) higher odds of HF compared to those in the bottom PRS decile.

## Pathway enrichment analysis recovers pathways relevant to HF
We used previously published and our newly identified HF GWAS variants ($n = 40$) together with the 18 proposed instruments for the ten MR-proteomics genes associated with HF and conducted gene pathway enrichment analysis using GTEx V8. These 58 variants are associated with 1605 GTEx V8 *cis*-eQTLs ($p < 1 \times 10^{-4}$), corresponding to a total of 165 unique genes (see Supplementary Data 12). After restricting the analysis to pathways described in Gene Ontology, KEGG, and Reactome, we observed 56 enriched pathways (FDR <5%). Biological pathways include muscle adaptation (adjusted $p$ value = 0.03), ventricular system development ($p = 0.03$), sarcomere organization ($p = 0.04$), regulation of vasculature development ($p = 0.04$), and aldosterone-regulated sodium reabsorption ($p = 0.04$), details on Supplementary Fig. 7 and Supplementary Data 13.

For the 18 GWAS distinct variants on HF, we determined the differential gene expression associated with the novel HF variants ($p < 1 \times 10^{-4}$) in each GTEx V8 tested tissue (heart atrial, heart ventricle, artery aorta, adipose, liver, kidney, and whole-blood tissues, and transformed cultured fibroblasts). We then used the set of differentially expressed genes to conduct an overrepresentation analysis on a per-tissue basis (Supplementary Fig. 8). A total of 605 enriched pathways had at least two differentially expressed genes, with heart-left ventricle being the tissue with the most significantly enriched pathways ($n = 393$). The rs6945340/*HIP1* variant showed the largest number of enriched pathways ($n = 391$, all tissues) with the heart's left ventricle being the primary tissue. Pathways to highlight for this variant include the Krebs cycle, respiratory electron transport chain (both with $p = 4.8 \times 10^{-30}$), and oxidative phosphorylation ($p = 3.2 \times 10^{-5}$). Further details are available in Supplementary Data 14. For eight of the MR-proteomics genes, we identified 77 reported associations with HF-related medical terms according to the EpiGraphDB database (Supplementary Data 15).

**Table 2 | Protein-hits for heart failure identified through Mendelian randomization that passed an FDR threshold of 5%**

| Protein Gene Name | Number of SNPs | Odds Ratio | 95% CI | P value | pHet† | FDR | MR-Egger Intercept (95% CI); P value | Coloc PP.H4* | Druggability classification (Chemical Modality) | Near known HF gene** |
|---|---|---|---|---|---|---|---|---|---|---|
| ITIH4* | 2 | 1.13 | (1.07, 1.17) | 2.51E-07 | 0.66 | 3.90E-04 | N/A | 0.97 | Non-druggable | No |
| APOC3* | 1 | 1.19 | (1.11, 1.28) | 1.74E-06 | NA | 1.35E-03 | N/A | 0.99 | Advanced Clinical Phase (Oligonucleotide) | No |
| MAPK3 | 3 | 0.95 | (0.93, 0.97) | 6.70E-06 | 0.41 | 3.48E-03 | 0.02 (−0.01, 0.05); 0.43 | 0.52 | Advanced Clinical Phase (Small molecule) | No |
| TNFSF12 | 2 | 0.96 | (0.94, 0.98) | 1.78E-05 | 0.05 | 6.94E-03 | N/A | 0.79 | Clinical Phase 1 (Antibody) | No |
| ABO | 2 | 1.02 | (1.01, 1.03) | 2.89E-05 | 0.11 | 8.99E-03 | N/A | 0.01 | Non-druggable | ABO |
| APOH* | 2 | 0.96 | (0.94, 0.98) | 5.24E-05 | 0.83 | 1.36E-02 | N/A | 0.89 | Non-druggable | No |
| B3GNT8 | 2 | 0.97 | (0.96, 0.99) | 9.35E-05 | 0.96 | 2.08E-02 | N/A | 0.48 | Non-druggable | No |
| NTN4 | 2 | 1.08 | (1.04, 1.13) | 1.10E-04 | 0.68 | 2.14E-02 | N/A | 0.04 | Non-druggable | No |
| DLL1 | 1 | 0.87 | (0.8, 0.93) | 1.53E-04 | NA | 2.65E-02 | N/A | 0.75 | Non-druggable | No |
| MST1 | 3 | 1.02 | (1.01, 1.03) | 1.99E-04 | 0.11 | 3.10E-02 | −0.20 (−0.38, −0.01); 0.29 | 0.37 | Non-druggable | No |
| ENPEP | 4 | 0.96 | (0.94, 0.98) | 3.12E-04 | 0.18 | 4.27E-02 | 0.01 (−0.02, 0.03); 0.62 | 0.74 | Non-druggable | PITX2, FAM241A |
| NAE1 | 1 | 0.82 | (0.74, 0.91) | 3.55E-04 | NA | 4.27E-02 | N/A | 0.6 | Advanced Clinical Phase (Small molecule) | No |
| TNXB | 1 | 1.03 | (1.02, 1.05) | 3.56E-04 | NA | 4.27E-02 | N/A | 0.61 | Non-druggable | No |
| SIRPA | 1 | 0.98 | (0.97, 0.99) | 3.94E-04 | NA | 4.39E-02 | N/A | 0.56 | Non-druggable | No |
| EBI3 | 1 | 0.75 | (0.64, 0.89) | 4.44E-04 | NA | 4.61E-02 | N/A | 0.01 | Non-druggable | No |
| IL27 | 1 | 0.75 | (0.64, 0.89) | 4.44E-04 | NA | 4.61E-02 | N/A | 0.4 | Non-druggable | No |

Gene names are italicized.

Significant MR results, FDR <5%. MR estimates were calculated using Wald ratio for instruments with one variant and inverse-variance weighting and fixed effects for instruments that contained more than one variant. Note that an OR >1 indicates an increase in protein corresponding with an increase in HF risk or vice versa, suggesting that the therapeutic solution may be an inhibitor; an OR <1 indicates either a decrease in protein levels corresponding with an increase in HF risk or an increase in protein levels corresponding with a decrease in HF risk, suggesting the therapeutic solution may be an agonist.

Genes that passed a colocalization threshold of *PP.H4* >0.5 (suggestive threshold) are highlighted in bold and *PP.H4* >0.8 (strong threshold) are marked with an asterisk.

MR Mendelian randomization, *FDR* false discovery rate, *PP.H4* posterior probability of H4.

*Posterior probability of H4 (one common causal variant) from colocalization of pQTL and GWAS results.

**Previously reported HF GWAS gene for instruments in GWAS loci (within 500 KB up or down from each loci).

†MR pHet were measured by Cochran's Q-test for heterogeneity across individual-variant MR estimates within a genetic instrument; instruments containing one variant were not tested for heterogeneity.

## Mouse knock-out models for novel genes identified by GWAS or MR-proteomics

We queried for knock-out (KO) mouse models, using the Mouse Genomics (MGI) resource, for evidence that modification of the target produces a phenotype relevant to HF. In 13 genes (eight GWAS and five MR-proteomics genes), we retrieved evidence of a KO associated with cardiovascular abnormalities. KO models on *CAMK2D*, *PRKD1*, *MAPK3*, *NAE1*, *SLC39A8*, *PHIP*, *RFX4*, *SCARB1*, and *TNXB* showed phenotypes such as myocardial abnormalities, dilated cardiomyopathy, abnormal response to cardiac infarction, and cardiac hypertrophy, suggesting an intrinsic role in heart function regulation (Supplementary Data 16).

## Druggability

A total of seven novel genes from the GWAS (*CAMK2D*, *PRKD1*, and *PRKD3*) and MR-proteomics (*MAPK3*, *TNFSF12*, *APOC3*, and *NAE1*) were identified to encode proteins that are predicted to be druggable (*CAMK2D*) or targets for 14 unique drugs that are either licensed or in the clinical phase (*PRKD1*, *PRKD3*, *MAPK3*, *TNFSF12*, *APOC3*, and *NAE1*). Except for drugs targeting Apolipoprotein C-III mRNA, Volanesorsen, and AKCEA-APO-CIII-L$_{Rx}$ evaluated for familial chylomicronemia syndrome, all the other 12 drugs are either licensed or under clinical investigation for cancer (*n* = 10 (*MAPK3*, *PRKD1*, *PRKD3*, and *NAE1*)) or autoimmune disorders (*n* = 2, (*TNFSF12*)). In four of the seven druggable genes, we were able to use our MR findings to infer the type of pharmacological action (agonist versus antagonist) needed to prevent HF and compared this against the pharmacological action of the existing drugs with a single target (which are most likely to reproduce genetic findings). Through this process, we observed a match in one gene (*APOC3*); and for the other druggable genes (*MAPK3*, *NAE1*, and *TNFSF12*), the existing drugs were an inhibitor/antagonist, while MR suggested an agonist, details on Supplementary Data 17.

## In silico trials

We searched for genetic associations for the GWAS hits and conducted two-sample MR for the MR proteomics hits to evaluate safety and efficacy outcomes relevant to the primary prevention trials on HF. Seven of the 18 GWAS distinct variants and two of the ten MR-proteomics genes were additionally associated ($p < 1 \times 10^{-4}$) with efficacy outcomes (CAD, T2DM) in the same direction as HF (Supplementary Data 18). None of the 18 distinct GWAS variants or ten MR-proteomics genes showed an association ($p < 1 \times 10^{-4}$) with the following safety traits: cancers (lung, prostate, colorectal, breast), chronic kidney disease, Alzheimer's disease, liver enzymes, or creatinine.

## Comparison with Global Biobank Meta-analysis Initiative (GBMI) on HF

An unpublished study from the GBMI reporting a multi-ancestry HF GWAS (68,408 HF cases and 1,286,331 controls) identified 11 potentially novel loci for HF[15]. We compared these associations with our HERMES-MVP GWAS and determined that seven of the 11 GBMI variants were associated ($p < 5 \times 10^{-8}$) in our HF meta-analysis. None of these variants were associated ($p < 5 \times 10^{-8}$) in the HF GWAS in MVP African-Americans dataset (Supplementary Data 19). Two GBMI loci correspond to the same variants (rs10455872/*PLG* and rs600038/*SURF1*) previously reported by HERMES or MVP, and an additional five loci were in LD ($r^2$ range: 0.39 to 1) with our findings (Supplementary Data 19). Finally, two GBMI GWAS variants (rs17035646 and rs61208973) showed suggestive evidence of association in our HF GWAS ($p < 0.003$). In a replication study of the 18 novel loci, findings from the HF GWAS in the GBMI multi-ancestry excluding UK Biobank indicate 33.3% (6 of 18) of variants are significant ($p$ value <0.05/18), 61.1% (11 of 18) are nominally significant ($p$ value <0.05), and 100% have a beta estimate that is directionally concordant with our meta-analysis (Supplementary Data 20).

## Discussion

Our genetic analysis on HF consisting of 90,653 cases identified 18 distinct HF variants through GWAS and an additional ten putatively causal genes for HF through MR and colocalization using proteomic instruments. Our study expands the knowledge on the biological pathways associated with all HF risk loci discovered to date and identifies seven druggable genes as potential drug targets for the primary prevention of HF.

We conducted several strategies to provide biological credibility to our 18 distinct GWAS variants. First, 16 of the 18 variants showed genetic associations with HF risk factors that were directionally concordant with the HF findings, and several LV cardiac MRI traits. Second, overrepresentation analysis using differentially expressed genes by each GWAS variant identified the heart LV myocardium as the most significantly enriched tissue and recovered several pathways of HF relevance. Third, systematic querying on KO mouse models identified *CAMK2D*, *PRKD1*, *PHIP*, *RFX4*, *SLC39A8*, and *SCARB1*, genes found by our GWAS, with phenotypes relevant to HF. Novel variants to highlight include rs3820888/*SPATS2L* and rs4755720/*HSD17B12* that showed associations with HF risk factors and LV cardiac MRI traits. The rs3820888/*SPATS2L* variant showed evidence of colocalization with six cardiac MRI traits, including LVEF, LV mass to end-diastolic volume ratio, and AF, all of which were directionally concordant with the HF findings. Previous GWAS have also indicated that the same variant was also associated with QT interval[16]. The rs4755720/*HSD17B12* variant colocalized with LV end-diastolic volume indexed to BSA and HF risk factors that were directionally concordant with the HF findings, all showing a protective effect. Previous GWAS indicated that this variant, as well as others in strong LD, associated with a reduction in adiposity measures and an increase in lung function metrics, suggesting that cardiometabolic fitness may explain the association with HF[17–19].

We conducted MR-proteomic analyses to uncover the putative causal role of human proteins in HF. Ten genes passed our genetic colocalization test, of which nine were also not in LD with a previously reported HF variant, minimizing the probability of confounding by LD. Seven of the 10 genes showed associations with at least one HF risk factor, and in the majority (71%) of these associations, the point estimate was directionally concordant with the MR findings on HF.

Four (*MAPK3*, *PRKD1*, *CAMK2D*, and *PRKD3*) of the seven druggable genes identified by our analyses encode proteins with serine/threonine kinase activity. These four genes are associated with HF risk factors in a manner that is concordant with the findings on HF. *CAMK2D* also showed a suggestive association ($p = 9 \times 10^{-4}$) with LV mass. In support of our findings, a mouse model with deletion of *MAPK3*/*MAPK1* genes developed cardiac hypertrophy and ventricular dilation followed by reduced ventricular performance[20]. *CAMK2D*, *PRKD1*, and *PRKD3* are calcium/calmodulin-dependent protein kinases known to be associated with cardiac pathophysiology. Protein kinase-D, encoded by the *PRKD1* gene, appears to be a regulator of myocardial structure and function. Mice with a deletion of *PRKD1* in cardiomyocytes were reported to be resistant to stress-induced hypertrophy in response to pressure overload, angiotensin-II, and adrenergic activation[21]. Calcium/Calmodulin-Dependent Protein Kinase II (CamKII) is composed of four chains, one of which, delta (δ), is encoded by the *CAMK2D* gene. CamKII-δ is largely expressed in cardiac tissue (confirmed by our pathway enrichment analysis), where it regulates proteins involved in calcium handling, excitation-contraction coupling, activation of hypertrophy, cell death, and inflammation[22]. Several case-control studies have shown an upregulation of cardiac CamKII-δ expression and activity in patients with HF, dilated cardiomyopathy, and diabetic cardiomyopathy. In support of this, several experimental studies in animal models of dilated cardiomyopathy and HF have shown that chemical inhibition of CamKII led to protection from cardiac dysfunction, adverse cardiac remodeling, and cardiac arrhythmias[22]. More recently, the administration of a novel ATP-

competitive CaMKII-δ oral inhibitor (RA306) in a dilated cardiomyopathy mouse model led to an improvement in ejection fraction[23]. This oral inhibitor offers the opportunity to test the causal role of CamKII-δ through clinical trials for the prevention of HF. Interestingly, the *CAMK2D* gene was also associated with AF, confirming an association demonstrated by in-vitro and animal models of AF[22].

Additional druggable genes identified were *APOC3*, *TNFSF12*, and *NAE1*. The *APOC3* gene, which achieved the highest level of evidence in our analyses (FDR 5% and PP.H4 >0.8), is known for its associations with lipids, and CAD, which were confirmed in our analysis. Apolipoprotein C-III mRNA is targeted by two different antisense oligonucleotides (ASO), Volanesorsen and AKCEA-APO-CIII-$L_{Rx}$, evaluated for familial chylomicronemia syndrome. Phase 3 trials on Volanesorsen have shown an increase in LDL-C levels and thrombocytopenia, which makes it an unlikely candidate for the prevention of HF.[24] AKCEA-APO-CIII-$L_{Rx}$ is an ASO liver-specific that appears to have a better safety profile, and may be more suitable for long-term use[25]. *TNFSF12* gene encodes for the TNF superfamily member 12 protein; increased levels of this protein were associated with a risk reduction in HF according to our MR and colocalization findings. Similar, directionally concordant, findings were reported by recent MR proteomics (using various proteomics platforms) against ischemic stroke[26]. These results are consistent with the finding that *TNFSF12* is MR associated and colocalized with AF, a risk factor for both ischemic stroke and HF. In addition, we observed a clear reduction in LV mass to end-diastolic volume ratio and a suggestive ($p = 2 \times 10^{-3}$) increase in LVEF, both directionally concordant with a risk reduction in HF. Transgenic mice and adenoviral-mediated gene expression models have also pointed to the role of *TNFSF12* in the development of dilated cardiomyopathy and severe cardiac dysfunction[27]. *NAE1* gene encodes NEDD8 activating enzyme E1 subunit 1 protein, and our MR and colocalization findings showed this gene was associated with lower values of blood pressure, which coincides with the reduced risk of HF.

The strengths of the current analysis are multiple. First, the large number of HF cases included in our analysis led us to identify new variants and putatively causal genes for HF through GWAS and MR proteomics. Second, we used three complementary strategies− nearest gene (local method), PoPs (global method), and pLoF− to assign the most likely gene responsible for the GWAS signal with HF. Through this process, we observed agreements in 11 of 18 GWAS variants, which provided some degree of confidence in the gene prioritization. However, we acknowledge that the PoPs method will miss variants that do not act through various mechanisms captured by PoPs[13], highlighting the challenge in assigning the gene responsible for GWAS loci[28–30]. Third, we provide biological credibility for most of our genetic findings through an extensive and complementary analysis covering HF risk factors, LV cardiac MRI, and -omics. Fourth, in seven MR hits for HF, we showed that our proposed instruments, in addition to associations with HF risk factors or LV cardiac MRI traits, were also associated with gene expression, and protein levels all acting in *cis*. Fifth, KO models of thirteen genes identified through GWAS and MR developed highly relevant phenotypes to HF and in some cases (CAMK2D), specific pharmacological inhibition showed reversibility of the HF phenotypes. Six, the lack of associations between the distinct GWAS loci and MR genes with safety outcomes used in the primary prevention trials of HF provides some reassurance on target safety profiles.

The degree of credibility on the causality of proteins identified by MR depends on whether the MR assumptions are valid. First, our colocalization analysis on HF, risk factors for HF, and LV cardiac MRI traits make confounding by LD unlikely. The selection of *cis*-variants as proposed instruments minimize the chances of horizontal pleiotropy. To further minimize the chances of horizontal pleiotropy, we developed a novel analysis that attempted to empirically test the relevant conditions needed for horizontal pleiotropy to invalidate MR. First, we looked for secondary proteins or gene expression associated with our

MR protein-hits, and then evaluated if those secondary proteins/gene expression were associated with HF and fall in a biological or PPI pathway outside our protein-hits. After doing this, only *TNXB* showed some evidence of horizontal pleiotropy. Interestingly, *cis*-pQTLs used as instruments for TNXB were not associated with *cis*-eQTLs, HF risk factors or LV cardiac MRI traits. Although we used multiple lines of evidence to determine putative causal genes, the pathway enrichment analysis identifies pathways linked to cardiac biology, but may not point to specific insights for HF, and we did not functionally validate any of our results, which remains as the highest level of evidence to support causal roles for the hits, especially those that pass the suggestive MR and coloc thresholds of FDR 5% and PP.H4 >0.5.

Although most of our variants and genes showed associations with HF risk factors that were biologically concordant with HF risk, some discordant associations were observed. HDL-C and diastolic BP accounted for most of these discordant associations. It has been reported that higher levels of diastolic BP may be protective on HF[31,32], instead of deleterious as we assumed, while the HDL-C association with HF seems to be non-linear[32], which was not accounted for in our MR analysis that included HDL-C as a co-variable. We validated seven of the 11 variants reported in an unpublished multi-ancestry HF GWAS by GBMI[15]. Another limitation is that our analysis was restricted to individuals of European ancestry. While this does reduce the potential bias caused by population stratification, our results may not apply to populations of other ancestral groups. Future HF GWAS meta-analysis including larger releases of MVP, All of US, and GBMI will not only provide chances for replication of variants identified in Europeans, but also include non-white populations to further increase the discovery of genetic determinants of HF.

Although the absence of HF subtypes in this analysis most certainly decreased our ability to detect signals specific to HF subtypes, it does not invalidate the ones identified. Evidence from primary prevention trials using HF as an outcome (as our genetic study) that uncovered the benefits of BP lowering therapies and statins indicates the plausibility for translation of our genetic findings. Future genomic analysis should extend to different HF subtypes, with a focus on HF with preserved ejection fraction, a major unmet need in medicine. Although our design attempted to emulate a primary prevention trial on HF, further studies with access to individual participant data that reliably recreate eligibility criteria and outcome ascertainment that cover efficacy (including HF subtypes) and safety outcomes are needed.

In conclusion, we discovered a total of 18 distinct novel HF-associated variants and ten putatively causal genes for HF through GWAS and MR-proteomics with evidence of biological plausibility. The new mechanisms and pathways together with the seven druggable genes discovered provide a tractable path for the translation of our genomic findings for the primary prevention of HF.

## Methods
### Clinical and demographic characteristics
The study population for the meta-analysis consisted of 1,279,610 participants, of which 302,287 were from MVP (43,344 cases and 258,943 controls) and 977,323 were from HERMES Consortium (47,309 cases and 930,014 controls). The clinical and demographic features of the participants are summarized in Supplementary Data 1. A detailed breakdown of clinical and demographic characteristics according to each study included in the HERMES Consortium has been previously published[8]. The population characteristics of the BioVU PRS cohort can be found in Supplementary Data 21.

### Genotyping, quality control, and imputation of genetic data
For the data obtained from the Million Veteran Program (MVP), DNA was extracted from participants' blood and genotyped using the MVP 1.0 Genotyping Array, which is enriched for both common and rare

genetic variants of clinical significance. Imputation performance was assessed, and variants that had poor quality as determined were removed from further analyses. All studies included in the HERMES Consortium utilized high-density genotyping arrays. A detailed table summarizing the genotyping, quality control, imputation, and analysis across the 29 distinct datasets included in the HERMES Consortium has been previously described[8]. For quality control, the per-variant call rate and the per-sample call rate across all studies was at least greater than 90[8]. The MAF threshold ranged from >0 to 1% across studies[8]. Further details can be found in the Supplementary Information.

## Phenotyping of heart failure
Across all 26 cohorts of the HERMES Consortium, cases with HF were identified by a clinical diagnosis of HF of any etiology, as determined by physician diagnosis or adjudication, ICD codes, and imaging, and controls were participants without a clinical diagnosis of HF. In the MVP, HF patients were identified as those with an International Classification of Diseases (ICD)−9 code of 428.x or ICD-10 code of I50.x *and* an echocardiogram performed within 6 months of diagnosis (median time period from diagnosis to echocardiography was 3 days, interquartile range 0−32 days). Further details can be found in the Supplementary Information.

## Genome-wide association study for HF
We performed a fixed effects inverse-variance weighted meta-analysis HF from the published MVP ($n = 302,258$) and HERMES ($n = 964,057$)[8] GWAS using METAL[33] (version release 2020-05-05) in a total of 1,266,315 individuals. We removed variants with a MAF <0.5%, resulting in 10,227,138 associations.

We used FUMA[34] to annotate our results using the default settings. In accordance with the default FUMA parameters, we defined distinct variants to have an $R^2 < 0.6$ and determined the associations that were >500 KB from a previously reported indexed variant in MVP and HERMES. We used the closest gene to the indexed variant and the top gene per locus identified by PoPs to prioritize genes for our GWA-significant ($p < 5 \times 10^{-8}$) loci.

The PoPS method[13] is a new gene prioritization method that identifies the causal genes by integrating GWAS summary statistics with gene expression, biological pathway, and predicted protein−protein interaction data. We applied the PoPS score because it has been shown to nominate causal genes at non-coding GWAS loci with greater predictive confidence compared to other similarity-based or locus-based methods[13]. By leveraging a framework unbiased by previous trait-specific knowledge, the PoPs tool can prioritize causal genes and therefore highlight relevant biological pathways with greater confidence. First, as part of the PoPS analysis, we used MAGMA to compute gene association statistics (z-scores) and gene−gene correlations from GWAS summary statistics and LD information from the 1000 Genomes. Next, PoPS performs marginal feature selection by using MAGMA to perform enrichment analysis for each gene feature separately. The model is fit by generalized least squares (GLS), and MAGMA results are used to perform marginal feature selection, retaining only features that pass a nominal significance threshold ($p < 0.05$). Then, PoPS computes a joint enrichment of all selected features simultaneously in a leave one chromosome out (LOCO) framework. The gene features employed by PoPS are listed here: https://github.com/FinucaneLab/gene_features. The PoPs method uses data from gene expression datasets, protein−protein interaction networks, and pathway databases; however, variants that act through mechanisms not captured by the PoPs model would not be identified. Finally, PoPS computes polygenic priority scores for each gene by fitting a joint model for the enrichment of all selected features. The PoP score for a gene is independent of the GWAS data on the chromosome where the gene is located. The PoPS analysis returned scores for a total of 18,383 genes per set of GWAS datasets. We then annotated

our GWAS loci with the Ensembl genes in a 500 kb window and selected the highest PoP score gene in the locus as the prioritized gene. For all the genes suggested by the nearest gene and PoPS, we conducted gene-burden tests derived using a gene-based (mean) approach in a mixed model framework using the Genebass-UK Biobank resource (see Supplementary Information).

## Genome-wide association study in African-Americans MVP subpopulation
We conducted a GWAS of HF in the African-American MVP subpopulation and performed lookups for our novel HF variants as well as the previously described HF variants. The African-American subpopulation in the MVP is composed of 11,399 cases with heart failure and 69,726 controls, of which 94.9% cases and 85.4% controls were male with a mean age of 63.82 (9.92) and 56.39 (12.20) for the cases and controls, respectively (Supplementary Data 4).

## Associations of HF GWAS variants with HF risk factors and LV cardiac MRI traits
For genetic variants that passed the GWAS threshold for HF ($p < 5 \times 10^{-8}$), we determined genetic associations for 15 HF risk factors and nine LV cardiac MRI traits derived from available GWAS. Data on HF risk factors was obtained from European-descent GWAS studies: BMI[35], smoking[36], alcohol intake frequency[37], AF[38], diastolic and systolic BP[39], T2DM[40], CAD[41], LDL-C[42], HDL-C[42], estimated glomerular filtration rate (eGFR)[29], and chronic obstructive airways disease (COPD)[36], and troponin I cardiac muscle, N-terminal proBNP (NT-proBNP), and interleukin-6 (IL-6).

For LV cardiac MRI traits, we determined genetic associations from two separate publications. Seven LV cardiac MRI measurements in 36,041 participants of the UK Biobank from ref. 43 and LV mass and LV mass to end-diastolic volume ratio from cardiac MRI in 42,157 UK Biobank participants from Aung et al. (unpublished) using automated CMR analysis techniques and LV GWAS techniques[44,45].

We used $p < 1 \times 10^{-4}$ (0.01/number of secondary to HF traits tested in the manuscript) to account for multiple testing. For associations that passed our $p$ value threshold, we evaluated whether the directionality of HF risk factors associations was concordant with findings on HF; for example, for a variant that showed an increased risk of HF, we expect a positive association with a deleterious risk factor.

## Mendelian randomization on 1557 proteins and HF
**Selection of proposed pQTL instruments.** We obtained pQTLs from a genome-proteome-wide association study in the Fenland study of 10,708 participants of European-descent[12] (retrieved from www.omicscience.org). The genome-proteome-wide association study was conducted using 10.2 million genetic variants and plasma abundances of 4775 distinct protein targets (proteins targeted by a least one aptamer) measured using the SOMAscan V4 assay[12]. Significant genetic variant pQTLs were defined as passing a Bonferroni $p$ value threshold of $p < 1.004 \times 10^{-11}$. Approximate conditional analysis was performed to detect secondary signals for each genomic region identified by distance-based clumping of association statistics[12]. To diminish the likelihood of horizontal pleiotropy, we restricted proposed instrumental variables to (lead and secondary signals) *cis*-pQTLs using a $p$ value threshold of $p < 5 \times 10^{-8}$ in marginal statistics, where *cis* is defined as any variant within a +/− 1 Mb region of the protein-encoding gene. A total of 2900 *cis*-pQTLs across 1557 genes (mean = 1.9, min = 1, max = 14) covering an equal number of proteins from the Fenland study were used as proposed instrument variables for conducting two-sample MR of proteomics against HF.

## Mendelian randomization and colocalization
We performed two-sample MR using the TwoSampleMR package in R (https://mrcieu.github.io/TwoSampleMR/)[46]. The Wald Ratio was

used for instruments with one variant and the inverse-variance weighted MR method was used for instruments with two or more variants. We tested the heterogeneity across variant-level MR estimates, using the Cochrane Q method (mr_heterogeneity option in TwoSampleMR package) and plotted the effects of the variants on the proteins against the effects of the variants on HF to validate our instruments when more than one variant was included. We defined significant MR results using a false discovery rate (FDR) of 0.05 calculated by the Benjamini–Hochberg method (corresponding $p$ value = $5 \times 10^{-4}$). We used the MR-Egger intercept test to detect potential directional pleiotropy, and report the Egger intercept and corresponding standard error and $p$ value for genes with three or more variants, where the MR-Egger intercept can be interpreted as an estimate of the average horizontal pleiotropic effect of the genetic variants[47].

MR assumes the SNP influences the outcome only through exposure. To help guard against the existence of distinct but correlated causal variants for the exposure and outcome, for results that passed our MR threshold (FDR <0.05), we performed colocalization using the COLOC package[48] in R. Colocalization assesses the probability of a shared causal variant (PP.H4) or distinct causal variants (PP.H3) between the HF GWAS and $cis$-pQTL instruments for the protein of interest. We performed conditional analysis on the pQTL data to identify conditionally distinct pQTL signals and performed colocalization using marginal (unadjusted) pQTL results as well as results conditional on each of the instruments used in the MR. Statistically significant MR hits with a posterior probability of a shared causal variant (PP.H4) >0.5 for at least one instrumental variant were then investigated further. Colocalization was performed using with default priors (prior probability of initial trait association is $1 \times 10^{-4}$, prior probability of shared causal variant across two traits is $1 \times 10^{-5}$). We also investigated if the $cis$-pQTL instruments for genes that passed both MR and colocalization thresholds were also $cis$-eQTLs ($p < 5 \times 10^{-8}$). Tissues used were whole blood from eQTLGen and heart atrial, heart ventricle, artery aorta, adipose, liver, kidney tissues, and transformed cultured fibroblasts from GTEx V8.

### MR and colocalization for HF risk factors and cardiac MRI traits

For proteins that passed both MR and colocalization thresholds, we conducted two-sample MR analyses of these proteins, using $cis$-pQTLs from the Fenland study as proposed instrumental variables, against 15 HF risk factors and nine cardiac MRI traits described in the previous section (see Supplementary Material for details on traits and datasets). For the MR results that passed a $p$ value threshold of $p < 1 \times 10^{-4}$, we conducted colocalization analyses as previously described. We defined significant findings as those that passed thresholds for MR ($p < 1 \times 10^{-4}$) and colocalization (PP.H4 >0.5).

### Assessment of horizontal pleiotropy

For statistical findings that passed the MR and colocalization thresholds, we evaluated the possibility that horizontal pleiotropy may invalidate our findings. The pipeline of analysis is depicted in Supplementary Fig. 5. Step-1: We determined if our $cis$-pQTLs were associated ($p < 5 \times 10^{-8}$) with other proteins levels included in SOMAscan V4 or with gene expression using data from eQTLGen. Step-2: We queried if the genes (including genes that encode SOMAscan proteins) identified in Step-1 were within 1MB of the risk loci for HF identified by GWAS conducted to date. Step-3: We conducted a two-sample MR to identify if the secondary genes/proteins (identified in Step-1) were associated with HF, using a Bonferroni-corrected $p$ value (0.05/number of unique genes/proteins identified in Step-1). We leveraged as proposed instruments the lead $cis$-pQTL ($p < 5 \times 10^{-8}$) from the Fenland study, and if it was not available, we used the lead $cis$-eQTL ($p < 5 \times 10^{-8}$) identified from eQTLGen. Step-4: We then mapped all secondary genes/proteins identified in Step-3 to Reactome/KEGG

pathways; and compared if these pathways are on the same (vertical pleiotropy) or different (horizontal pleiotropy) pathway as that associated with the primary genes identified through MR proteomics for HF. To further investigate the physiological functionalities of our findings retrieved in Step-4, we queried two databases: the Enrichr[49–51], an interactive gene knowledge discovery database, and the GPS-Prot server[52], a platform with aggregated information about protein–protein interactions.

### LD score regression

We used LD Score regression[53] (LDSC) to estimate genetic correlations between heart failure and 15 cardiovascular traits. We estimated using European LD scores obtained from the 1000 Genomes Project Phase 3 data for the HapMap2 SNPs. We used MungeSumstats to perform standardization of association statistics[54].

### Polygenic risk score analysis

A polygenic score for heart failure was calculated using the HF meta-analysis using the PRS-CS package[55], which utilizes a Bayesian regression framework to calculate posterior SNP effect sizes under a continuous shrinkage prior. We used the LD reference panel constructed using the 1000 Genomes Project Phase 3 data. We conducted these analyses in Python, using the packages scipy and h5py. The PRS was evaluated in the Vanderbilt University Medical Center (VUMC) BioVU, a biobank that links the de-identified electronic medical record (EMR) system containing phenotypic data to discarded blood samples from routine clinical testing for the extraction of genetic data[56]. A full description of the BioVU resource has been previously published[56]. Participants with heart failure were identified by a modified version of the eMERGE definition for heart failure, which includes the *International Classification of Diseases*, Tenth Revision (ICD-10) codes, where age was defined as age at heart failure for cases and age at last medical visit for controls. To determine the ability of PRS to stratify heart failure cases from controls, we used a logistic regression model, adjusting for age, sex, and three principal components of ancestry in the BioVU. We assessed enrichment in the more extreme tail of the PRS distribution by evaluating the odds ratio for individuals in the top PRS decile compared to individuals in the bottom PRS decile. In the top decile of PRS, there were 723 participants with HF and 6788 controls, and in the bottom decile, there were 416 participants with HF and 7096 controls.

### Pathway enrichment analysis

We conducted an enrichment analysis to identify biological pathways associated with HF risk loci (established and novel) that passed the GWAS $p$ value thresholds. For each locus, we selected the top variant and then identified $cis$-eQTLs (within a 1 Mb region) from GTEx V8 in any tissue associated with the top variants and extracted all genes with a $p < 1 \times 10^{-4}$. We merged all retrieved genes to a gene set that was then used for inquiry for the enriched pathways. This set of genes was set forth to an overrepresentation analysis using the pathways described in Gene Ontology, KEGG, and Reactome. Selected pathways were those significantly enriched at an FDR <0.05.

Additionally, we explored the downstream transcriptional consequences associated with the distinct variants identified by our GWAS on HF and those not previously reported. We used the distinct variants and conducted a differential gene expression analysis (using a dominant model) for all transcripts available in GTEx V8 for heart atrial, heart ventricle, artery aorta, adipose, liver, kidney, transformed cultured fibroblasts, and whole-blood tissues. After fitting models for our variants, we retrieved all genes differentially expressed at a $p < 1 \times 10^{-4}$ and conducted an enrichment pathway analysis (through an overrepresentation analysis, as described above). Enrichment analyses were performed using the R packages clusterProfiler and enrichplot[57].

## EpiGraphDB queries

To investigate the current knowledge about the biomedical functions of the hit genes in association with HF, we used the EpiGraphDB database[58]. We queried the biomedical and epidemiological relationships curated in the database to identify associations between the genes we identified and cardiovascular-related outcomes and risk factors (see Supplementary Methods).

## Querying the MGI database

We queried the Mouse Genome Informatics (MGI, http://www.informatics.jax.org/) resource for all candidate genes from our novel GWAS hits list or those suggested as causal from our MR/colocalization approach. MGI uses a standardized nomenclature, and controlled vocabularies such as the Mouse Developmental Anatomy Ontology, the Mammalian Phenotype Ontology, and the Gene Ontologies. As MGI extracts and organizes data from the primary literature, we have parsed all system abnormalities associated with models on all of the queried genes[59]. For models that displayed cardiovascular abnormalities, we have hand-curated the abnormalities and organized them into three distinct groups associated with (1) congenital heart malformations, (2) myocardial abnormalities, and (3) vascular abnormalities.

## Druggability annotations

Proteins encoded by genes identified in the GWAS and MR analyses for HF were annotated with drug tractability information based on information provided by OpenTargets[10,60,61] (release 2021-03-08). Open-Targets tractability system stratified drug targets into nine mutually exclusive groups (termed "buckets") based on the drug type and the stage of the drug discovery pipeline. For easier interpretation, we regrouped the original buckets into four mutually exclusive groups, as follows: Licensed drugs: bucket-1 for antibodies, small molecules, and other modalities. Drugs in clinical development: buckets 2 and 3 for antibodies, small molecules, and other modalities. Compounds in the preclinical phase: buckets 4 and 5 for small molecules. Predicted druggable: buckets 6 to 8 for small molecules plus buckets 4 and 5 for antibodies. The remaining proteins were considered non-druggable. For genes that were the target of licensed drugs, we checked whether the disease indication was also a risk factor for HF, as this may introduce a bias analogous to confounding by indication in MR.

## GBMI replication of novel loci

We conducted a replication of the 18 novel loci in the Global Biobank Meta-analysis Initiative (GBMI) multi-ancestry GWAS on heart failure, which includes 859,141 controls and 60,605 cases from BioBank Japan, BioMe, BioVU, China Kadoorie Biobank, Estonian Biobank, FinnGen, Genes & Health, HUNT, Lifelines, Michigan Genomics Initiative, Partners Biobank, UCLA Precision Health Biobank, excluding UK Biobank[62]. Heart failure cases were ascertained by ICD code (phecode 428.2). We consider $p < 0.05/18$ as a level of significance for replication and $p < 0.05$ as a level of nominal significance.

## Reporting summary

Further information on research design is available in the Nature Portfolio Reporting Summary linked to this article.

## Data availability

The MVP GWAS summary statistics used in this study is available through dbGAP under accession code phs001672.v10. The only restriction is that use of the data is limited to health/medical/biomedical purposes, and does not include the study of population origins or ancestry. Use of the data does include methods development research (e.g., development and testing of software or algorithms) and requesters agree to make the results of studies using the data available to the larger scientific community. The HERMES GWAS summary statistics used in this study are publicly available in the GWAS Catalog under accession code GCST009541. Fenland-SomaLogic protein GWAS data are available at https://omicscience.org/. GTEx project v.8 data were publicly available at https://gtexportal.org/home/. Mouse Genome Informatics (MGI) data is publicly available at http://www.informatics.jax.org/. The GWAS summary statistics for the risk factor analyses used in this study are deposited in the GWAS Catalog (https://www.ebi.ac.uk/gwas/) and the accession codes are as follows: body mass index (GCST006900), alcohol consumption (GCST007325), atrial fibrillation (GCST006414), systolic blood pressure (GCST006624), diastolic blood pressure (GCST006630), type-2 diabetes (GCST006867), and coronary artery disease (GCST005194) troponin (GCST005806), NT-pBNP (GCST005806) and IL-6 (GCST90012049). The GWAS summary statistics for smoking and chronic obstructive airways disease used in this study are available at https://gwas.mrcieu.ac.uk under GWAS ID ukb-b-5779 and ukb-b-13447, respectively, and the GWAS summary statistics for the traits examined in the in silico trials are available at https://gwas.mrcieu.ac.uk using the GWAS IDs listed in the Supplementary Data. The GWAS summary statistics for the LDL-cholesterol and HDL-cholesterol are publicly available at http://csg.sph.umich.edu/willer/public/glgc-lipids2021/results/ancestry_specific/. The summary statistics for estimated glomerular filtration rate (eGFR) are deposited in://www.uni-regensburg.de/medizin/epidemiologie-praeventivmedizin/genetische-epidemiologie/gwas-summary-statistics/index.html. The cardiac MRI datasets provided by Pirruccello et al. are deposited under Dataset Name "UK Biobank Cardiac MRI LV GWAS" on https://cvd.hugeamp.org/downloads.html. The Open Targets data are deposited in https://platform.opentargets.org/. The EpiGraphDB database used in this study is provided at: https://www.epigraphdb.org/.

## Code availability

We used publicly available software for the analyses, and all software used is listed and described in the Methods section of our manuscript. Statistical analyses were conducted in R version 3.6.3. Mendelian randomization analyses were conducted using the TwoSampleMR package in R version 0.5.3 (https://mrcieu.github.io/TwoSampleMR/), genetic colocalization analyses were conducted using the coloc package in R (https://cran.r-project.org/web/packages/coloc/index.html and https://chr1swallace.github.io/coloc, using default priors), pathway enrichment analyses were conducted using the clusterProfiler package in R (https://pubmed.ncbi.nlm.nih.gov/22455463/) and the enrichplot R package, LD Score regression was conducting using LDSC (https://github.com/bulik/ldsc), and polygenic risk score was calculated using the PRS-cs package v1.0.0 (https://github.com/getian107/PRScs). Meta-analysis of GWAS summary statistics were prepared using publicly available software, including METAL (https://genome.sph.umich.edu/wiki/METAL_Documentation), version release 2020-05-05. The software used to annotate our results are described in the Methods section of the manuscript.

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

## Acknowledgements

We are grateful to all the MVP investigators; a list of MVP investigators can be found in Supplementary Information. This research is supported by funding from the Department of Veterans Affairs Office of Research and Development, Million Veteran Program Grant I01 CX001737 (PI: Phillips), and I01-BX004821 (PI: Wilson/Cho). This publication does not represent the views of the Department of Veterans Affairs or the United States Government. We also acknowledge the VA Merit Grant I01-CX001025 (PI: Wilson/Cho).

The Fenland study was approved by the National Health Service (NHS) Health Research Authority Research Ethics Committee (NRES Committee—East of England Cambridge Central, ref. 04/Q0108/19), and all participants provided written informed consent. We are grateful to all Fenland volunteers and to the General Practitioners and practice staff for assistance with recruitment. We thank the Fenland Study Investigators, Fenland Study Co-ordination team, and the Epidemiology Field, Data and Laboratory teams. The Fenland Study (10.22025/2017.10.101.00001) is funded by the Medical Research Council (MC_UU_12015/1). We further acknowledge support for genomics from the Medical Research Council (MC_PC_13046). Proteomic measurements were supported and governed by a collaboration agreement between the University of Cambridge and SomaLogic.

P.B.M. and S.E.P. acknowledge the support of the National Institute for Health and Care Research Barts Biomedical Research Centre (NIHR203330); a delivery partnership of Barts Health NHS Trust, Queen Mary University of London, St George's University Hospitals NHS Foundation Trust and St George's University of London. N.A. acknowledges support from the NIHR Integrated Academic Training program which supports his Academic Clinical Lectureship post. C.G. has received funding from the European Union's Horizon 2020 research and innovation program under the Marie Skłodowska-Curie grant agreement No 754490—MINDED project.

L.S.P. is supported in part by VA awards CSP #2008, I01 CX001899, I01 CX001737, and I01 BX005831; NIH awards R01 DK127083, R21 AI156161, UL1 TR002378, and U18DP006711; and a Cystic Fibrosis Foundation award PHILLI12A0. The sponsors had no role in the design and conduct of the study; collection, management, analysis, and interpretation of the data; and preparation, review, or approval of the manuscript. L.S.P. is also supported in part by the Veterans Health Administration (VA). This work is not intended to reflect the official opinion of the VA or the US government.

J.P.C. moved to work with Novartis Institute for Biomedical Research during the submission of this project.

## Author contributions

J.P.C. conceived the study design, oversaw all analyses and interpretations, and wrote the manuscript. J.P.C., J.J., Y.V.S., and C.La. conceived of the project. D.R., G.M.P., A.C.P., H.D., C.G., and B.R.F. performed the formal analyses and visualizations, and wrote the manuscript. E.W., N.A., M.P., and Q.H. contributed data. E.H.F.-E. and Q.S.W. contributed data. E.H.F.-E. performed analysis. N.M.K. contributed to project administration. J.W. edited the manuscript. L.G., D.C.P., A.P.B., C.Li., K.A., Z.W., B.C., J.E.H., P.W.F.W., L.S.P., P.B.M., S.E.P., K.C., A.R.L., M.P.M., and J.M.G. participated in the contribution of data or analysis tools. All authors critically reviewed the manuscript.

## Competing interests

The authors declare no competing interests.

## Additional information

[1]Division of Aging, Brigham and Women's Hospital, Harvard Medical School, 75 Francis St., Boston, MA 02130, USA. [2]Massachusetts Veterans Epidemiology Research and Information Center (MAVERIC), VA Boston Healthcare System, 150. S. Huntington Ave, Boston, MA 02130, USA. [3]Department of Biostatistics, Boston University School of Public Health, 801 Massachusetts Ave Crosstown Centre, Boston, MA 02118, USA. [4]Laboratory of Genetics and Molecular Cardiology, Heart Institute, University of São Paulo, Av Dr Eneas de Carvalho Aguiar 54, São Paulo 5403000, Brazil. [5]Genetics Department, Harvard Medical School, Harvard University, 77 Avenue Louis Pasteur, Boston, MA 02115, USA. [6]Broad Institute of MIT and Harvard, 415 Main St., Cambridge, MA 02142, USA. [7]Health Data Science Centre, Human Technopole, V.le Rita Levi-Montalcini, 1, Milan 20157, Italy. [8]Central RNA Lab, Non-coding RNAs and RNA-based Therapeutics, Istituto Italiano di Tecnologia, Via Morego 30, 16163 Genova, Italy. [9]MRC Epidemiology Unit, Institute of Metabolic Science, University of Cambridge, Addenbrookes Hospital, IMS, Box 285, Cambridge CB2 0QQ, UK. [10]William Harvey Research Institute, Barts and The London School of Medicine and Dentistry, Queen Mary University of London, London EC1M 6BQ, UK. [11]Barts Heart Centre, St Bartholomew's Hospital, Barts Health NHS Trust, West Smithfield, London, UK. [12]Computational Medicine, Berlin Institute of Health (BIH) at Charité – Universitätsmedizin Berlin, Kapelle Ufer 2, Berlin 10117, Germany. [13]Precision Healthcare University Research Institute, Queen Mary University of London, London, UK. [14]Vanderbilt Institute for Clinical and Translational Research, Vanderbilt University Medical Center, Nashville, TN, USA. [15]Vanderbilt University Med. Ctr., Departments of Medicine (Cardiology), Biomedical Informatics, and Pharmacology, Nashville, TN, USA. [16]BHF Cardiovascular Epidemiology Unit, Department of Public Health and Primary Care, University of Cambridge, Worts Causeway, Cambridge CB1 8RN, UK. [17]Department of Chemical Biology, European Molecular Biology Laboratory, European Bioinformatics Institute, Wellcome Genome Campus, Hinxton CB10 1SD, UK. [18]Department of Epidemiology, Emory University Rollins School of Public Health, 1518 Clifton Rd NE, Atlanta, GA 30322, USA. [19]Atlanta VA Health Care System, 1670 Clairmont Road, Decatur, GA 30033, USA. [20]Massachusetts General Hospital, Boston, MA 02114, USA. [21]Division of Cardiology, Department of Medicine, Emory University School of Medicine, 1639 Pierce Dr NE, Atlanta, GA 30322, USA. [22]Division of Endocrinology, Emory University, 101 Woodruff Circle, WMRB 1027, Atlanta, GA 30322, USA. [23]MRC Biostatistics Unit, University of Cambridge, Cambridge CB2 0SR, United Kingdom. [24]William Harvey Research Institute, Barts and The London Faculty of Medicine and Dentistry, Queen Mary University of London, Charterhouse Square, London EC1M 6BQ, UK. [25]National Institute for Health Research, Barts Biomedical Research Centre, Queen Mary University of London, London, UK. [26]William Harvey Research Institute, NIHR Barts Biomedical Research Centre, Queen Mary University of London, Charterhouse Square, London EC1M 68Q, UK. [27]Department of Biomedical Informatics, Emory University School of Medicine, 1639 Pierce Dr NE, Atlanta, GA 30332, USA. [28]Cardiology Section, VA Providence Healthcare System, 830 Chalkstone Avenue, Providence, RI 02908, USA. [29]Department of Medicine, Warren Alpert Medical School of Brown University, 222 Richmond Street, Providence, RI 02903, USA. [30]These authors contributed equally: Danielle Rasooly, Gina M. Peloso. [31]These authors jointly supervised this work: Claudia Langenberg, Yan V. Sun, Jacob Joseph, Juan P. Casas.
✉e-mail: drasooly@bwh.harvard.edu; jacob.joseph@va.gov

## VA Million Veteran Program

**Jennifer E. Huffman** [2], **Peter W. F. Wilson**[19,21], **Lawrence S. Phillips**[19,22], **Kelly Cho**[1,2], **John Michael Gaziano**[1,2], **Yan V. Sun** [18,19,27,31], **Jacob Joseph**[28,29,31] ✉ **& Juan P. Casas**[1,2,31]

A full list of members and their affiliations appears in the Supplementary Information.

