## [Peer Review File · Nature Communications]

Genome-wide association analysis and Mendelian randomization proteomics identify drug targets for heart failureEditorial Note: This manuscript has been previously reviewed at another journal that is not operating a transparent peer review scheme. This document only contains reviewer comments and rebuttal letters for versions considered at *Nature Communications* .

REVIEWER COMMENTS

Reviewer #2 (Remarks to the Author):

The authors can be commended for their additional revisions. In particular, the flow of the manuscript and therefore the readability have strongly improved. There remain some minor points.

First, and as previously pointed out by Reviewer #3, the coloc threshold of $PP4 > 0.5$ is relatively liberal, especially if the significance cutoff for the MR analysis was FDR 5% (rather than Bonferroni). I appreciate the effort of the authors to now also highlight the results with $PP4 > 0.8$, however I was surprised with how this is handled later on in the discussion, ie all results ($PP4 > 0.5$ and $PP4 > 0.8$) are discussed as though being of equivalent evidence. This is especially true as only a small number of the hits turn out to pass the standard $PP4 > 0.8$! I would suggest that the limitations of the liberal FDR5% + $PP4 > 0.5$ thresholds be mentioned in the MR discussion section, and that it is explicitly mentioned that functional validation and/or clinical trials would remain the highest level of evidence to support causal roles for the hits (especially those just passing FDR5% and $PP4 > 0.5$). Perhaps those with highest evidence be explicitly mentioned separately in the discussion also, for clarity to the readers.

Second, the replication results are highly appreciated, and I think these make an important contribution to the paper. However, the replication results (given large sample size) are somewhat underwhelming, though generally reasonable in terms of P-values. However, the signs seem a bit puzzling to me. How come ~60% of signals replicate at nominal $P < 0.05$, but only 50% have the same sign? That would suggest that some signals are replicating in terms of significance but are actually DISCORDANT in direction? Furthermore, in my personal experience, even in highly underpowered replication studies, the concordance in direction of effect can be quite high (>80-90%) even if almost no hits formally replicate. In this case, the 50% concordance in sign is perfectly in line with the null hypothesis.... (ie, null SNPs will have ~50% chance of having the same sign in a different dataset). Can the authors comment on this? Are you sure no errors were introduced in the evaluation of the betas/signs?

Reviewer #3 (Remarks to the Author):

The authors have satisfactorily responded to my comments and toned down the language and emphasized the critical limitations of the study.

April 2023

We thank Reviewer #2 for the thoughtful comments, and have addressed both. We have provided detailed point-by-point responses to the concerns and a manuscript with tracked changes.

Reviewer #2 (Remarks to the Author):

The authors can be commended for their additional revisions. In particular, the flow of the manuscript and therefore the readability have strongly improved. There remain some minor points.

First, and as previously pointed out by Reviewer #3, the coloc threshold of $PP4 > 0.5$ is relatively liberal, especially if the significance cutoff for the MR analysis was FDR 5% (rather than Bonferroni). I appreciate the effort of the authors to now also highlight the results with $PP4 > 0.8$, however I was surprised with how this is handled later on in the discussion, ie all results ($PP4 > 0.5$ and $PP4 > 0.8$) are discussed as though being of equivalent evidence. This is especially true as only a small number of the hits turn out to pass the standard $PP4 > 0.8$! I would suggest that the limitations of the liberal FDR5% + $PP4 > 0.5$ thresholds be mentioned in the MR discussion section, and that it is explicitly mentioned that functional validation and/or clinical trials would remain the highest level of evidence to support causal roles for the hits (especially those just passing FDR5% and $PP4 > 0.5$). Perhaps those with highest evidence be explicitly mentioned separately in the discussion also, for clarity to the readers.

Response. We thank the Reviewer for the comment, and we agree. We have added the limitation in the Discussion:

“Although we used multiple lines of evidence to determine putative causal genes, the pathway enrichment analysis identifies pathways linked to cardiac biology, but may not point to specific insights for HF, and we did not functionally validate any of our results, which remains as the highest level of evidence to support causal roles for the hits, especially those that pass the suggestive MR and coloc thresholds of FDR 5% and $PP.H4 > 0.5$.” [DISCUSSION]

We have also explicitly mentioned in the Discussion the hits with the highest level of evidence.

“The APOC3 gene, which achieved the highest level of evidence in our analyses (FDR 5% and $PP.H4 > 0.8$), is known for its associations with lipids, and CAD, which were confirmed in our analysis.” [DISCUSSION]

Second, the replication results are highly appreciated, and I think these make an important contribution to the paper. However, the replication results (given large sample size) are somewhat underwhelming, though generally reasonable in terms of P-values. However, the

signs seem a bit puzzling to me. How come ~60% of signals replicate at nominal $P < 0.05$, but only 50% have the same sign? That would suggest that some signals are replicating in terms of significance but are actually DISCORDANT in direction? Furthermore, in my personal experience, even in highly underpowered replication studies, the concordance in direction of effect can be quite high (>80-90%) even if almost no hits formally replicate. In this case, the 50% concordance in sign is perfectly in line with the null hypothesis.... (ie, null SNPs will have ~50% chance of having the same sign in a different dataset). Can the authors comment on this? Are you sure no errors were introduced in the evaluation of the betas/signs?

Response. We thank you very much for the suggestion to conduct the replication analysis, and thank you for identifying this. We had introduced an error in the evaluation of the concordance of the betas/signs, and have fixed this. The replication results are 100% concordant in direction. We have fixed this in the manuscript and the column titled "Beta estimate directional concordance" in Table S20. We have reviewed the entirety of the replication analysis; no changes were made to the other columns of this table.

"Findings from the HF GWAS in the GBMI multi-ancestry excluding UK Biobank indicate 33.3% (6 of 18) of variants are significant (p -value $< 0.05/18$), 61.1% (11 of 18) are nominally significant (p -value < 0.05), and 100% have a beta estimate that is directionally concordant with our meta-analysis (Table S20)." [RESULTS]